# Assessment of the Effects of the Herbicide Aclonifen and Its Soil Metabolites on Soil and Aquatic Environments

**Marioara Nicoleta Caraba** [1,2], **Diana Larisa Roman** [1,2], **Ion Valeriu Caraba** [3,*] **and Adriana Isvoran** [1,2]

[1] Department Biology-Chemistry, Faculty of Chemistry-Biology-Geography, West University of Timisoara, Pestalozzi 16, 300315 Timisoara, Romania; marioara.filimon@e-uvt.ro (M.N.C.); diana.roman@e-uvt.ro (D.L.R.); adriana.isvoran@e-uvt.ro (A.I.)

[2] Advanced Environmental Research Laboratories (AERL), Oituz 4, 300086 Timisoara, Romania

[3] Faculty of Bioengineering of Animal Resources, University of Life Sciences "King Mihai I" from Timisoara, Calea Aradului 119, 300645 Timisoara, Romania

* Correspondence: caraba_i@animalsci-tm.ro

**Abstract:** Aclonifen is an herbicide with a global market that is expected to grow in the period 2021–2027. This study focuses on revealing the effects of both aclonifen and of its soil metabolites on aqueous and soil environments. The soil's physicochemical parameters and activities of enzymes found in the soil (urease, dehydrogenase, catalase, alkaline phosphatase) were evaluated in laboratory conditions for soil treated with various doses of aclonifen (from half of a normal dose, 2.1 g/ha, to three times the normal dose) every 7 days for a period of 28 days. A computational approach was used to calculate the acute toxicity of aclonifen and of its soil metabolites on aqueous organisms and to evaluate the possible interactions of these compounds with soil enzymes. The results obtained in the experimental study showed a significant inhibitory effect of the herbicide aclonifen on the investigated enzymes; those activities were not recovered after 28 days in the presence of high doses of aclonifen. Statistically significant effects of the herbicide dose and exposure duration on enzymes' activities were observed using post hoc analysis at the $p < 0.001$ level. Pearson's correlations revealed that the investigated enzyme activities were usually strongly ($p < 0.001$) influenced by the organic matter, available phosphorus content, and nitrogen level. The outcomes of the computational study revealed moderate acute aqueous toxicity of aclonifen and of its metabolites. All metabolites were also able to bind to the investigated enzymes and may have an inhibitory effect on their activities.

**Keywords:** enzymatic activity; molecular docking; aquatic toxicity

## 1. Introduction

Pesticides are widely used in actual crop management to obtain more and cheaper food by protecting crops from pests and diseases. The global consumption of pesticides constantly grew in the last decade, arriving at approximately 2.7 million metric tons, meaning a 57% increase compared with that in 1990 [1]. Despite their beneficial effects regarding crop protection and agricultural efficiency, pesticides can affect the environment by leading to soil, water, and air pollution, may contaminate food, and may affect human health. Consequently, the evaluation of the potential risks of the use of pesticides for human health and ecological effects is one of the main concerns nowadays.

Due to their effects on the environment, the use of pesticides is strongly regulated. The Environmental Protection Agency (EPA) in the United States (http://npic.orst.edu/reg/laws.html---accessed on 24 May 2023) and the European Commission (https://food.ec.europa.eu/plants/pesticides---accessed on 24 May 2023) in the European Union have the authority to regulate pesticide use.

Among the pesticides, the group of herbicides is the most used in food production and crop management [2]. The excessive use of herbicides usually leads to negative impacts on both the environment and human health [3]. In the soil, herbicides are degraded, being used

as food especially by worms and soil microorganisms. The degradation of the herbicides in soil is due to several mechanisms involving chemical hydrolysis, microbial degradation, photolysis, volatility, and leaching. The combination of the physicochemical properties of the herbicide, the characteristics of soil, the environmental conditions, and crop practices strongly determine the degree of contribution of every of the above mentioned mechanisms involved in the degradation of herbicides [4]. Several groups of microorganisms found in soil are capable of degrading pesticides, with fungi (such as Aspergillus, Absidia, Fusarium, and Oxysporum) and bacteria (such as Rhodococcus, Pseudomonas, Flavobacterium, Azotobacter, and Acetobacter) being the groups mainly involved into the degradation and mineralization of pesticides through enzymatic reactions [5,6]. The major enzymes involved in pesticide degradation are hydrolases, oxidoreductases, esterases, phosphotriestarases, and glutathione S-transferases [5,7]. Through degradation in soil, pesticides can form metabolites that may have greater toxicity than the parent compound. Many ecological risk assessments for pesticides focus only on the effects of the parent compounds, overlooking the potential impacts of metabolites.

The determination of activities of specific enzymes found in soil can lead to a better understanding of the metabolic processes contributing to the biogeochemical cycles of nutrients, as soil pesticides can disrupt local metabolisms and implicitly affect the activities of enzymes [8,9]. Enzymatic activities are considered as indicators of the quality of soil because the decrease in the activities of soil enzymes due to contamination with various chemicals is faster compared with the changes in the physicochemical properties of the contaminated soil [10]. The activities of the enzymes found in soil are influenced, besides the pesticides applied, by the characteristics and physicochemical parameters of the soil that have a role in influencing the activity of microorganisms when interacting with pesticides: soil type, temperature, pH, the amount of humus, the content of nitrites, nitrates, ammonium, sulfur, phosphorus, conductivity, etc. [11–14]. The most frequently assessed activities for soil enzymes includes those of urease, dehydrogenase, phosphatase, proteases, and catalase [15–19]. The inhibitory effects of different groups of pesticides on the soil environment and, respectively, on the activities of enzymes found in soil have been reported in numerous studies [15,20–29]. The data presented in these studies are evidence that the assessment of the activity of enzymes found in soil can provide valuable information about the effects of pesticides on the soil environment.

The aquatic environment can be contaminated due to the application of pesticides through different routes: runoff, spray drift, and leaching. Contamination of aquatic ecosystems with pesticides can lead to health risks at all levels of biological organization, i.e., microorganisms, invertebrates, fish, and aquatic plants [30]. In aquatic ecosystems, pesticides may cause numerous effects such as lesions, reproductive failure, suppression of the immune system, and even mortality of the aquatic organisms [31]. Consequently, evaluation of the effects of pesticides on aquatic ecosystems is also of a great interest nowadays.

This study focuses on aclonifen (IUPAC name 2-chloro-6-nitro-3-phenoxyaniline), a small and hydrophobic diphenyl ether herbicide (MW = 264.66 Da, logP = 3.8) used in the pre-emergence control of weeds in agricultural crops. Aclonifen is considered to act on weeds by inhibiting two enzymes involved in the synthesis of chlorophyll and carotenoids [32] and is approved to be used in all the EU countries [33]. Data available in the Pesticide Properties Database reveal that aclonifen is moderately persistent in the soil with degradation rates in the field conditions of $DT_{50}$ = 80.4 days and $DT_{90}$ = 314 days, and that it produces a moderate effect on soil earthworms with acute $LC_{50}$ = 150 mg kg$^{-1}$ for *Eisenia foetida* at 14 days [33]. Aclonifen is considered to have only a small adverse effect on carbon mineralization in soil (from 8% for sand soil to 22% for loam soil) and to not have a significant adverse effect on nitrogen mineralization. Furthermore, aclonifen is considered as being stable in aqueous environments having a pH from five to nine, both by aqueous photolysis ($DT_{50}$ = 197 days) and aqueous hydrolysis, but it reveals a fast degradation in water sediment phase ($DT_{50}$ = 14.5 days). In the case of acute exposure, aclonifen is known to produce a moderate effect ($LC_{50}$ = 0.67 mg L$^{-1}$) on fish (*Oncorhynchus mykiss*) and on

aquatic invertebrates (*Daphnia magna*, $EC_{50}$ = 1.2 mg $L^{-1}$), but a strong negative effect on aquatic plants (*Lemna gibba*, $EC_{50}$ = 0.006 mg $L^{-1}$) [33].

Known soil metabolites of aclonifen are: 3-amino-2-chloro-4-nitrophenol, 3-chloro-4-phenoxybenzene-1,2-diamine, 4-(3-amino-2-chloro-4-nitrophenoxy)phenol, and 3,4-diamino-2-chlorophenol [33].

The aclonifen market is expected to raise at an annual growth rate of 6.48% for the years 2021–2027 [34]. Consequently, assessments of its effects on soil and aquatic environments are more than necessary. From the experimental point of view, the activity of several enzymes (urease, dehydrogenase, catalase, and phosphatase) used as soil quality indicators and the physicochemical properties of soil were assessed over a period of 28 days after the application of various doses of aclonifen in laboratory conditions. Computationally, the effects of the aclonifen and of its known soil metabolites on aquatic organisms were predicted and the molecular docking approach was used to evaluate the interactions of both aclonifen and of its metabolites with enzymes whose activities were evaluated experimentally.

The purpose, and at the same time the novelty of this study among others in the same field, consists of assessing the effects of aclonifen and of its soil metabolites on the activities of enzymes found in soil and on the aquatic organisms.

## 2. Materials and Methods

### 2.1. Materials

The herbicide under consideration for this study was aclonifen (2-chloro-6-nitro-3-phenoxyaniline, $C_{12}H_9ClN_2O_3$). Experiments were performed using "Proclus" (Bayer CropScience), a commercial product that is a suspension containing 600 g/L aclonifen.

For the computational study, the SMILES (Simplified Molecular Input Line Entry System) formulas were retrieved from PubChem database [35]. The same database was used to extract the 3D structures of aclonifen and its known soil metabolites. Figure 1 reveals the 2D formulas of aclonifen and its soil metabolites.

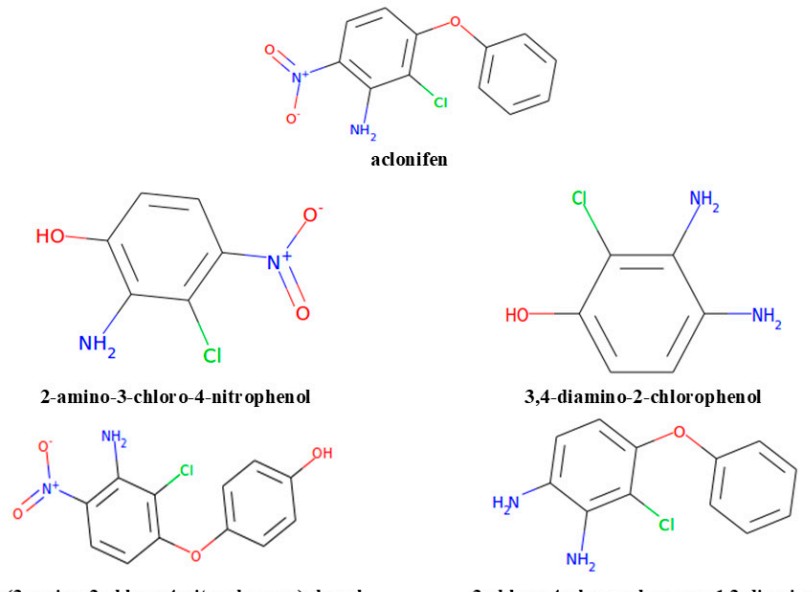

**Figure 1.** 2D formulas for aclonifen and for its known soil metabolites.

The SMILES formulas and the 3D structures of aclonifen and its soil metabolites were further used for computational purposes.

### 2.2. Soil Sampling and Treatment

The soil samples were collected from a field located in Ghiroda (Timis county, Romania, 45°45′14.54″ N, 21°18′16.66″ E), in an area where no pesticides or chemical fertilizers have ever been used. Soil samples (Chernozem) were collected from the top soil layer

(0–25 cm) from five different sites in amounts ranging from 30 to 35 kg. The soil was ground and sieved (2 mm) and then scooped using random sampling, yielding sub-samples of 10 kg. Samples were refrigerated and processed as soon as possible within the next 30 days. Solutions of aclonifen with established concentrations for each experimental variant (Table 1) were uniformly applied on the soil surface.

**Table 1.** Experimental variants considered in the present study. D—recommended dose.

| Experimental Variants | Dose | Explanation |
|---|---|---|
| M | 0 | control soil (soil not treated with aclonifen) |
| V1 | D/2 | 2.1 g aclonifen/ha |
| V2 | D | 4.2 g aclonifen/ha |
| V3 | 2×D | 8.4 g aclonifen/ha |
| V4 | 3×D | 12.6 g aclonifen/ha |

The amount of solution applied was determined according to the soil surface in each experimental variant to ensure the appropriate dose. The soil samples treated with aclonifen were incubated for 28 days in laboratory conditions.

### 2.3. Monitoring of the Physicochemical Properties of the Soil

Regarding soil characteristics, we monitored the following physicochemical parameters: temperature t (°C), electrical conductivity (EC), pH, content of organic matter (OM), water, ammonium ($NH_4$-N), nitrate-nitrogen ($NO_3$-N), and phosphate ($PO_4^{3-}$) in soil. Temperature, EC, and pH were measured by using a handheld multimeter (Multi 340i/SET WTW, Weilheim, Germany) that was fitted with a specific sensor for every parameter. To measure soil pH, soil samples were extracted with double distilled water (1:1 water to soil suspension). To measure the soil EC, a 5:1 suspension of double-distilled soil water was obtained. To determine the water content of the soil, the thermogravimetric analysis of the samples was carried out with the help of the Sartorius thermobalance, which allows the monitoring of the dehydration of the evaluated products via a constant weighing during the dehydration process. A quantity of 5 g of each investigated soil sample was oven-dried at 105 °C to constant weight using the thermo-gravimetric method [36]. The determination of OM in soil was carried out using the calcination method, and the loss of weight by combustion was determined by weighing. The soil samples were calcined in a muffle furnace, where the temperature was gradually increased to 550 °C and held constant for 4 h. The amount of organic matter for each soil sample was calculated as: $H\% = 100$, where $m_1$ is the weight of the soil sample at room temperature and $m_2$ is the weight of the ash sample [37]. Determination of $NH_4$-N, of phosphate, and of $NO_3$-N contents in each of the samples was performed by measuring the absorbance using a T90 UV/Vis spectrophotometer (PG Instruments, UK) at a wavelength of 630 nm for the $NH_4$-N [38], of 882 nm for the phosphate [39], and of 543 nm for the $NO_3$-N [40], respectively. These physicochemical properties were determined every 7 days for 28 days.

### 2.4. Biochemical Analyses

The enzyme activities considered in this study were: urease (EC 3.5.1.5), dehydrogenase (EC 1.1.1.1), alkaline phosphatase (EC 3.1.3.1) and catalase (EC 1.11.1.6). The activities of urease, dehydrogenase, and phosphatase were determined by using a T90 UV/Vis spectrophotometer (PG Instruments Ltd., Leicestershire, UK), and the titration method was used to determine the catalase activity. For each sample, the protocol was performed in triplicate, in controlled laboratory conditions, on the same day, by the same researcher. Dehydrogenase activity (DHA) was measured using 2,3,5-triphenyltetrazolium chloride (TTC) as substrate and the reaction product (triphenylformazane, TPF), monitored at 485 nm. The reaction mixture containing 3 g soil sample, 0.5 mL of TTC solution of 3%, and 1.2 mL Tris buffer (0.1 M, pH 7.6) was kept at 37 °C for 48 h. TPF was extracted with 20 mL acetone

and the absorbance of the supernatant was measured at 485 nm. The DHA was expressed as mg TPF g$^{-1}$ soil for 48 h [41].

Urease activity (UA) evaluated the rate of urea decomposition in ammonia ($NH_3$) and carbon dioxide ($CO_2$). For each sample, 5 g of soil was mixed with 5 mL phosphate buffer, 2 mL toluene, and 5mL of 5% urea solution ($CH_4ON_2$) in a sterile polyethylene tube. This mixture was incubated for 24 h at 37 °C. The quantity of produced $NH_4^+$ in the collected supernatant was determined using Nessler's reagent. The absorbance was measured at 445 nm, with the UA being expressed as mg $NH_3$-N g$^{-1}$h$^{-1}$ soil for 24 h [14,42].

Phosphatase activity (PhA) was estimated based on hydrolytic cleavage of phenyl phosphate by phosphomonoesterases with the final products being phenol and disodic phosphate. Phenol reacted with Gibbs reactive and a blue precipitate resulted. For each sample, about 2.5 g soil was mixed with a solution of 10 mL of 0.5% disodic phosphate. The mixture was incubated at 37 °C for 48 h. Following this, 50 mL of ammonium aluminum sulfate ($NH_4Al(SO_4)_2 \times 12H_2O$) was added to each solution and the new mixture was filtered through ash-free filter paper. 1 mL filtrate was further transferred to an empty test tube and mixed with 5 mL Borax solution ($Na_2B_4O_7 \times 10H_20$, pH = 9.4)) and bi-distilled water until a volume of 25 mL was reached. PhA was determined at 597 nm and expressed as mg phenol g$^{-1}$ soil for 48 h [19]. Catalase activity (CA) was obtained using the permanganometric method [43]. 3 g soil, 2 mL $H_2O_2$ 3%, and 10 mL phosphate buffer were mixed and incubated at 37 °C for 1 h. CA was expressed as the amount of oxygenated water that was not decomposed in 1 g of soil The activities of these enzymes were determined every 7 days for a period of 28 days.

### 2.5. Predictions of the Effects of Aclonifen and of Its Soil Metabolites on Aquatic Organisms

The Toxicity Estimation Software Tool (TEST) version 5.1.2 was used to estimate the toxicity of aclonifen and of its soil metabolites to aquatic organisms [44]. TEST uses QSAR (Quantitative Structure Activity Relationship) methodologies for predicting the following toxicity endpoints for aquatic organisms: the 96 h *Fathead minnow* 50 percent lethal concentration (LC$_{50FM}$), the 48 h *Daphnia magna* 50 percent lethal concentration (LC$_{50DM}$), and the *Tetrahymena pyriformis* 50 percent growth inhibition concentration (IGC$_{50}$).

### 2.6. Molecular Docking Study

Aclonifen and its known metabolites from soil were docked with structures of enzymes for which the effect of aclonifen on their activity was experimentally evaluated: urease, dehydrogenase, phosphatase, and catalase. The structures of these enzymes were extracted from the Protein Data Bank (PDB) [45]. They were chosen such as to correspond to enzymes belonging to microorganisms found in soil in complexes with substrates or inhibitors (when available) and having the highest resolution among the existent crystallographic structures: 1KEV chain A for *Clostridium beijerinckii* dehydrogenase [46], 4AC7 chain C for *Bacillus pasteurii* urease [47], and 4I9C chain A for *Bacillus subtilis* phosphatase [48]. In the case of catalase, the AlphaFold structural model AF-P42234-F1 for *Bacillus subtilis* catalase 2 was used [49]. In order to locate the active site of the catalase 2, the AlphaFold model was superposed to the structure of *Exiguobacterium oxidotolerans* catalase and a very good match was obtained for the two structures: RMSD = 0.851 Å for 410 carbon alpha pairs from the 480 total pairs (Figure S1). The structures of aclonifen, of its known soil metabolites, and those of the enzymes were prepared for molecular docking by means of the DockPrep utility under Chimera software [50]. The molecular docking was implemented using SwissDock online server [51]; a blind and accurate docking was selected. The docking outcomes were analyzed using Chimera.

### 2.7. Statistical Analysis

For each enzyme activity (DHA, UA, PhA, CA), the data sets corresponding to each combination of two independent variables (in this case, herbicide dose and exposure time) were checked for normality and homogeneity of variance (homoscedasticity) using

Anderson–Darling and Levene tests, respectively [52]. For enzyme activities fulfilling both these requirements, two-way ANOVA tests [53] were run, considering the herbicide dose and exposure time as independent variables. Post hoc analysis was performed using the Bonferroni procedure for each significant main effect. All pairwise comparisons for herbicide dose and exposure time were conducted against controls (herbicide dose as main effect) and the earliest time point (exposure duration as main effect). For significant interactions between herbicide dose and exposure time, these comparisons were made for each time point using the corresponding controls as reference groups. Finally, Pearson's correlations [54] were applied to the cumulative mean values for enzyme activities and to the measured values for selected soil physicochemical parameters (temperature, pH, OM, levels of nitrogen, ammonia, and phosphorus). Statistical analyses were performed using Statistica software, version 7 (StatSoft Inc., Tulsa, OK, USA). Statistical significance was defined at a two-tailed *p* value below 0.05.

## 3. Results

### 3.1. Assessment of the Effects of Aclonifen on Enzymatic Activities

Average values for the measured enzymatic activities for the soil samples treated with various doses of aclonifen after 7, 14, 21, and, respectively, 28 days of incubations are shown in Table 2. Data sets for these variables were normally distributed ($p \geq 0.076$). The tests for homogeneity of variance were not significant irrespective of enzymatic activity analyses ($p \geq 0.098$). The assumptions underlying the application of the two-way ANOVA were therefore fulfilled for both factors, herbicide dose and exposure time.

**Table 2.** Enzymatic activities in soil samples treated with various doses of aclonifen: DHA—dehydrogenase activity, UA—urease activity, CA—catalase activity, PhA—phosphatase activity. Marked boxes indicate significant differences as compared with the reference group (Bonferroni test, *** $p < 0.001$). The meaning of the experimental variants is given in Table 1.

| Time | Experimental Variant | DHA mg TPF g$^{-1}$ Soil | UA mg NH$_3$-N g$^{-1}$h$^{-1}$ Soil | CA mg H$_2$O$_2$ Undecomposed g$^{-1}$ Soil | PhA mg Phenol g$^{-1}$ Soil |
|---|---|---|---|---|---|
| | M | 3.95 ± 0.09 | 1.45 ± 0.04 | 6.48 ± 0.16 | 2.58 ± 0.07 |
| | V1 | 3.75 ± 0.04 | 0.90 ± 0.02 *** | 6.11 ± 0.13 *** | 2.30 ± 0.26 |
| 7 days | V2 | 2.73 ± 0.04 *** | 1.04 ± 0.04 *** | 5.91 ± 0.06 *** | 2.43 ± 0.14 |
| | V3 | 2.52 ± 0.07 *** | 0.86 ± 0.01 *** | 4.46 ± 0.05 *** | 2.29 ± 0.03 |
| | V4 | 1.11 ± 0.03 *** | 0.62 ± 0.05 *** | 4.21 ± 0.07 *** | 2.03 ± 0.07 *** |
| | M | 3.10 ± 0.16 | 1.28 ± 0.03 | 3.05 ± 0.12 | 2.14 ± 0.15 |
| | V1 | 3.20 ± 0.19 | 1.11 ± 0.03 | 2.96 ± 0.02 | 2.64 ± 0.08 *** |
| 14 days | V2 | 1.68 ± 0.04 *** | 0.95 ± 0.07 *** | 2.38 ± 0.04 *** | 2.13 ± 0.14 |
| | V3 | 1.61 ± 0.06 *** | 0.86 ± 0.06 *** | 2.30 ± 0.02 *** | 2.12 ± 0.03 |
| | V4 | 0.39 ± 0.00 *** | 0.78 ± 0.03 *** | 2.21 ± 0.09 *** | 1.92 ± 0.05 |
| | M | 2.49 ± 0.09 | 2.48 ± 0.02 | 4.63 ± 0.14 | 2.14 ± 0.00 |
| | V1 | 1.16 ± 0.05 *** | 1.50 ± 0.03 *** | 5.20 ± 0.08 *** | 2.07 ± 0.01 |
| 21 days | V2 | 0.62 ± 0.06 *** | 1.08 ± 0.02 *** | 4.21 ± 0.06 *** | 1.99 ± 0.06 |
| | V3 | 0.51 ± 0.06 *** | 0.85 ± 0.07 *** | 4.14 ± 0.05 *** | 2.07 ± 0.09 |
| | V4 | 0.33 ± 0.06 *** | 0.95 ± 0.05 *** | 3.28 ± 0.02 *** | 1.52 ± 0.09 *** |
| | M | 2.21 ± 0.08 | 2.49 ± 0.11 | 1.27 ± 0.01 | 2.01 ± 0.05 |
| | V1 | 1.54 ± 0.03 *** | 1.43 ± 0.11 *** | 1.13 ± 0.01 *** | 1.59 ± 0.07 *** |
| 28 days | V2 | 1.26 ± 0.03 *** | 1.06 ± 0.01 *** | 0.95 ± 0.01 *** | 1.38 ± 0.01 *** |
| | V3 | 1.13 ± 0.01 *** | 0.82 ± 0.06 *** | 0.54 ± 0.01 *** | 1.19 ± 0.01 *** |
| | V4 | 0.78 ± 0.06 *** | 0.58 ± 0.05 *** | 0.38 ± 0.01 *** | 0.89 ± 0.10 *** |

Figure 2 reveals the evolution of the dehydrogenase and urease activities depending on the applied dose of aclonifen (Figure 2a and, respectively, Figure 2c) and on the incubation time (Figure 2b and, respectively, Figure 2d). Similarly, Figure 3 emphasizes the evolution of enzymatic activities of phosphatase and catalase depending on the applied dose of

aclonifen (Figure 3a and, respectively, Figure 3c) and on the incubation time (Figure 3b and, respectively, Figure 3d).

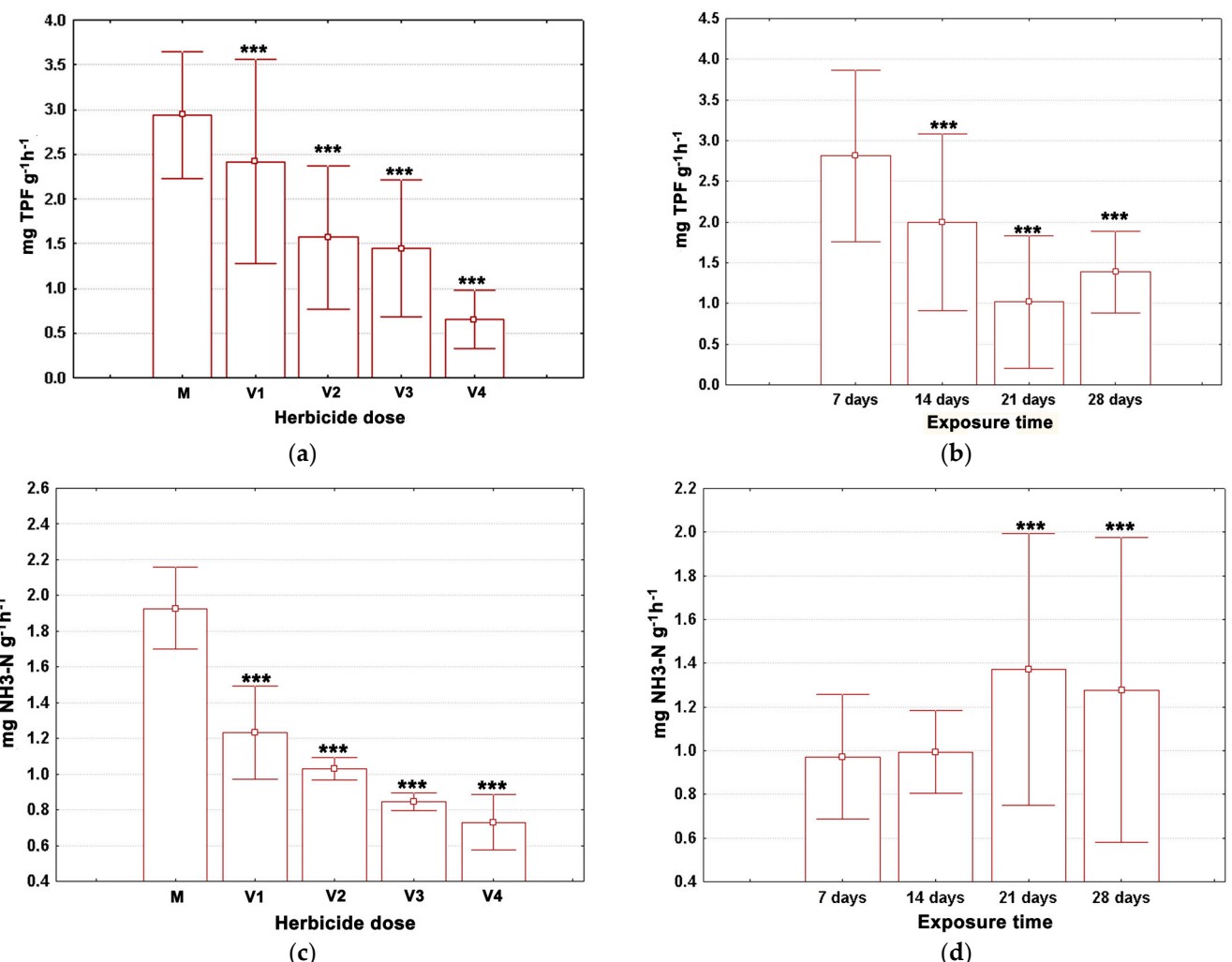

**Figure 2.** Evolution of enzymatic activities depending on the applied dose of aclonifen and on incubation time: dehydrogenase activity dependence on the dose of aclonifen (**a**) and on the incubation time (**b**); urease activity dependence on the dose of aclonifen (**c**) and on the incubation time (**d**). Parametric statistics was used to analyze data related to dehydrogenase activity and urease activity. The measured values for dehydrogenase activity and urease activity are shown as mean (box) with one standard deviation (error bar). Marked boxes indicate significant differences as compared with the reference group (Bonferroni test, *** $p < 0.001$; for phosphatase activity: Dunn's test, *** $p < 0.001$).

### 3.2. Correlations between the Enzymatic Activities and Physicochemical Properties of the Soil

The physicochemical parameters of the soil are shown in Supplementary Table S1. The Pearson correlation was applied to the cumulative mean values for enzyme activities and the values measured for the investigated soil physicochemical parameters (temperature, pH, humidity, OM, nitrogen level, ammonia level, and available phosphorus). Significant correlations were noticed between the investigated enzymatic activities and the analyzed soil properties (Table 3).

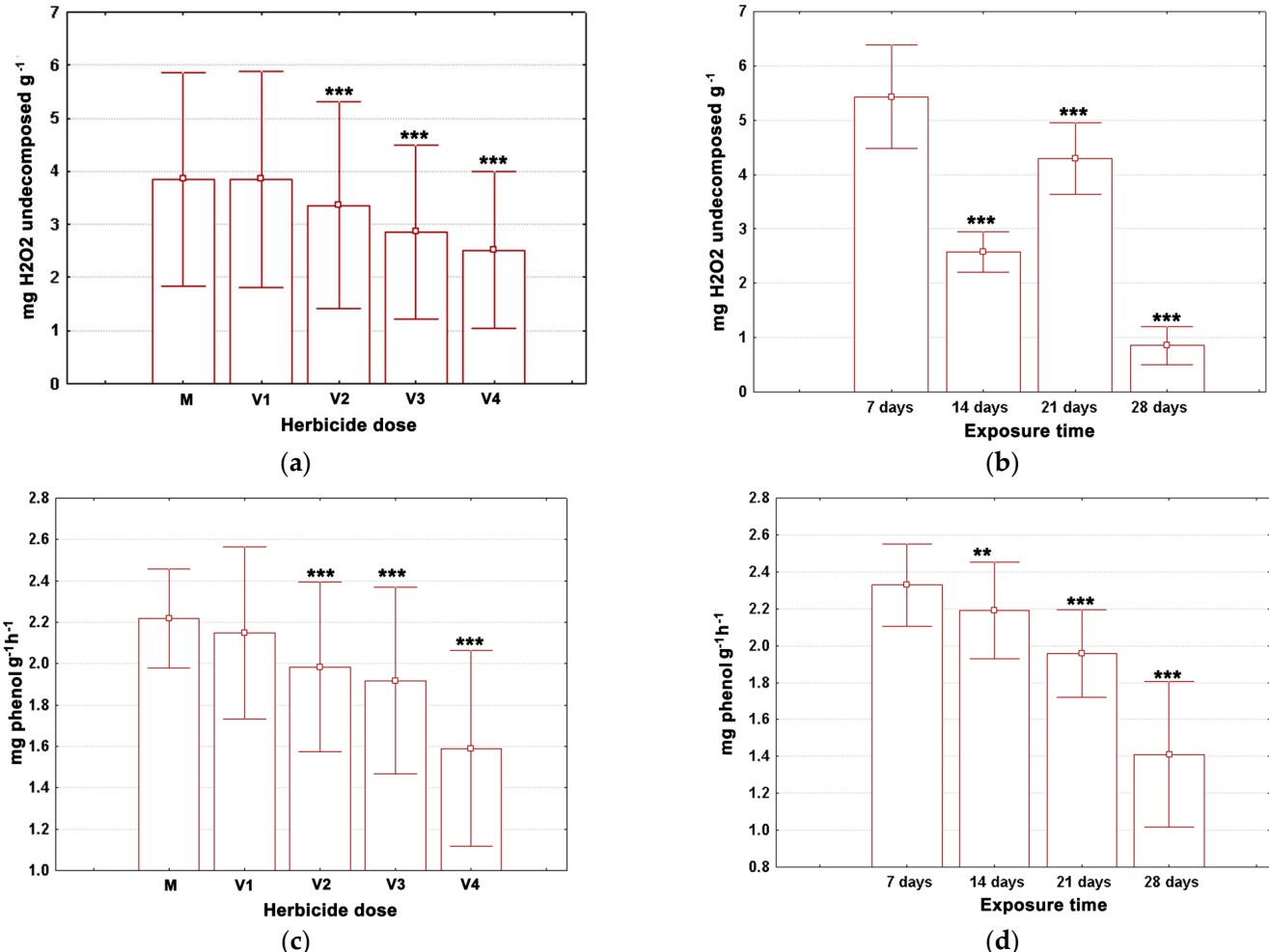

**Figure 3.** Evolution of enzymatic activities depending on the applied dose of aclonifen and on incubation time: catalase activity dependence on the dose of aclonifen (**a**) and on the incubation time (**b**); phosphatase activity dependence the dose of aclonifen (**c**) and on the incubation time (**d**). Parametric statistics were used to analyze data related to catalase activity and phosphatase activity. The measured values for catalase activity and phosphatase activity are shown as mean (box) and one standard deviation (error bar). Marked boxes indicate significant differences as compared with the reference group (Bonferroni test, *** $p < 0.001$, ** $p < 0.01$; for phosphatase activity: Dunn's test, *** $p < 0.001$, ** $p < 0.01$).

**Table 3.** Correlations between the enzymatic activities and the soil physicochemical parameters: DHA—dehydrogenase activity, UA—urease activity, CA—catalase activity, PhA—phosphatase activity, EC—electric conductivity, OM—organic matter content. Marked boxes indicate significant correlations (*** $p < 0.001$, ** $p < 0.01$, * $p < 0.05$).

| Physicochemical Parameters/Enzymatic Activity | DHA | UA | CA | PhA |
|---|---|---|---|---|
| pH | 0.46 * | 0.42 | 0.10 | 0.27 |
| EC | 0.20 | 0.42 | 0.01 | 0.41 |
| Soil moisture | 0.13 | 0.28 | 0.10 | 0.27 |
| OM (mg/g) | 0.71 *** | 0.40 | 0.73 *** | 0.53 * |
| N-NO$_3$ (mg/g) | 0.35 | 0.45 * | 0.23 | 0.28 |
| N-NH$_4$ (mg/kg) | 0.66 *** | −0.08 | 0.58 ** | 0.32 |
| Available phosphorus (mg/kg) | 0.49 * | 0.22 | 0.49 * | 0.49 * |

### 3.3. Predicted Toxicity of Aclonifen and of Its Soil Metabolites against Aquatic Organisms

The results obtained using TEST software and concerning the acute toxicity of aclonifen and of its soil metabolites against aquatic organisms are presented in Table 4.

**Table 4.** Acute toxicity of aclonifen and of its soil metabolites against aquatic organisms. FM—*Fathead minnow*, DM—*Daphnia magna*, TP—*Tetrahymena pyriformis*.

| Chemical Compound | $LC_{50}$FM 96 h (mg/L) | $LC_{50}$DM 48 h (mg/L) | $IGC_{50}$TP 48 h (mg/L) |
|---|---|---|---|
| aclonifen | 1.03 | 0.87 | 2.49 |
| 2-amino-3-chloro-4-nitrophenol | 19.56 | 1.47 | 12.97 |
| 3,4-diamino-2-chlorophenol | 73.3 | 2.32 | 83.22 |
| 4-(3-amino-2-chloro-4-nitrophenoxy)phenol | 1.04 | 0.82 | 2.10 |
| 3-chloro-4-phenoxybenzene-1,2-diamine | 3.54 | 0.93 | 7.97 |

### 3.4. Computational Assessment of the Interactions of Aclonifen and of Its Soil Metabolites with Enzymes Found in Soil

The results of the molecular docking study are shown in Figure 4 for the interactions between aclonifen and dehydrogenase, phosphatase, catalase, and urease. The binding energies for the interactions of the aclonifen and its soil metabolites with the investigated enzymes are presented in Table 5.

**Table 5.** Interaction energies for the interactions of aclonifen and its soil metabolites with the investigated soil enzymes. It should be noted that neither aclonifen nor any of its soil metabolites bind to the active site of *Bacillus pasteurii* urease.

| Chemical Compound/Interaction Energy | $\Delta$G (kcal/mol) | | |
|---|---|---|---|
| | Dehydrogenase | Phosphatase | Catalase 2 |
| aclonifen | −7.50 | −6.81 | −7.60 |
| 2-amino-3-chloro-4-nitrophenol | −6.43 | −5.95 | −6.72 |
| 3,4-diamino-2-chlorophenol | −6.75 | −6.42 | −7.17 |
| 4-(3-amino-2-chloro-4-nitrophenoxy)phenol | −7.53 | −6.76 | −7.72 |
| 3-chloro-4-phenoxybenzene-1,2-diamine | −7.28 | −6.95 | −7.45 |

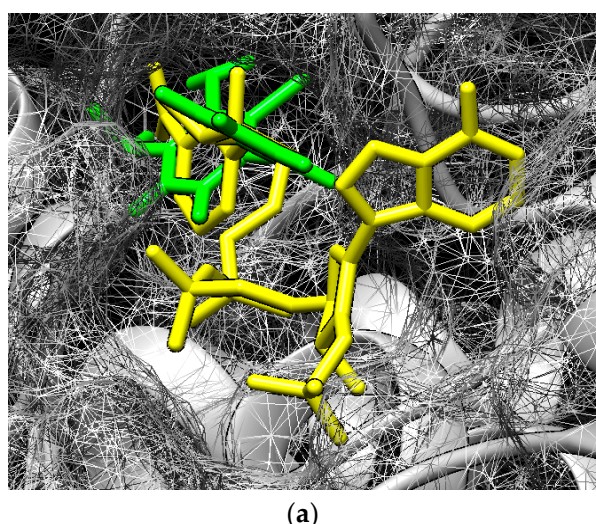

(**a**)                                                   (**b**)

**Figure 4.** *Cont.*

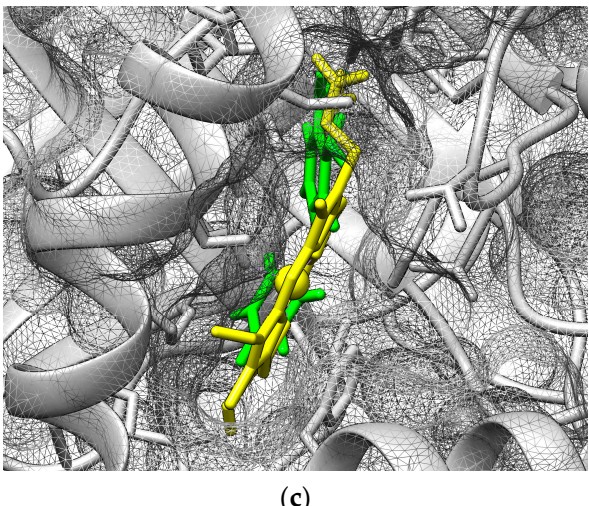 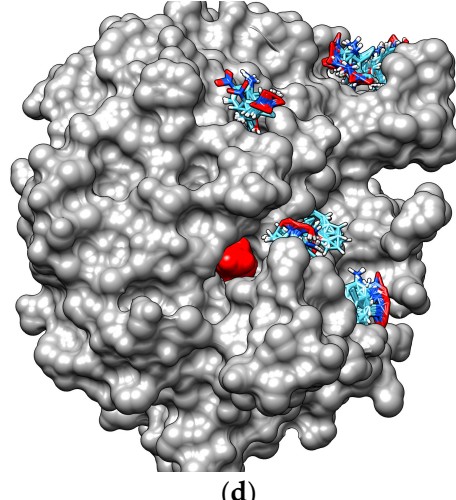

(**c**)                                                      (**d**)

**Figure 4.** Molecular docking outcomes regarding the interactions of aclonifen with enzymes found in soil: (**a**) dehydrogenase—gray mesh surface, NADPH—yellow sticks, aclonifen—green sticks; (**b**) phosphatase—gray mesh surface, PHR signaling peptide—yellow sticks, aclonifen—green sticks; (**c**) catalase—gray mesh surface, hem—yellow sticks, aclonifen—green sticks; (**d**) urease—gray solid surface, citrate—red solid surface, aclonifen—sticks colored by atom type (carbon—light blue, oxygen—red, nitrogen—dark blue, hydrogen—white, chloride—green).

## 4. Discussion

Data presented in Table 2 and Figures 2 and 3 reveal the information presented in the following. The herbicide dose had a significant effect on DHA (F (4, 59) = 1643.83, $p < 0.001$, η2 = 0.523). Post hoc analyses with the Bonferroni procedure revealed a significant dose-dependent diminution in the measured values (Figure 2a). The effect of exposure duration on DHA was also significant (F (3, 59) = 1595.34, $p < 0.001$, η2 = 0.381) (Figure 2b), but its relative impact was 30% lower as compared with that of herbicide dose. The implication of this effect was interpreted as a significant consequence of the interaction between herbicide dose and exposure time (F (12, 59) = 94.22, $p < 0.001$, η2 = 0.091). This means that the main effect of exposure duration depended on the level of herbicide used. After the exposure of 7 days, the obtained values for the DHA were significantly lower in the experimental variants V2, V3, and V4 compared with those in the unexposed control, M (Table 2). A similar trend was observed after 14 days. The decrease in DHA values was even stronger at 21 and 28 days (Table 2), with all treatment groups showing significantly reduced DHA compared with the control (M). These data indicate a significant decreasing trend for all 28 days in DHA for the experimental variants starting at the normal dose. This outcome emphasizes a particular susceptibility of dehydrogenase to the presence of aclonifen in the soil, in good correlation with published data, which usually reveals a negative impact of herbicides on DHA regardless of application dose and soil properties [14,16,17,25]. Dehydrogenases only occur in viable cells and, consequently, their activity is strictly correlated with the microbial community of soil. It was observed that DHA exhibits an increasing trend under anaerobic conditions [55]. Accordingly, the inhibition of DHA suggests that the anaerobic microbial community was decreased in the presence of aclonifen and that oxygen diffusion was increased. Similarly, a significant mean effect of the herbicide dose on UA (F (4, 59) = 869.98, $p < 0.001$, η2 = 0.682) was recorded. The post hoc comparisons showed a significant dose-dependent UA decrease (Figure 2c). There also was a significant main effect of exposure duration on UA (F (3, 59) = 202.53, $p < 0.001$, η2 = 0.115). The relative impact of exposure time on UA was six-fold lower than that seen for the herbicide dose. With respect to the first time point (Figure 1d), the UA values were similar after 14 days but increased significantly after 21 days and remained significantly elevated at 28 days (Figure 2d). There also was a statistically significant interaction between the herbicide dose and the exposure duration (F (12, 59) = 84.88, $p < 0.001$, η2 = 0.019). After 7 days (Table 2), the measured UA

levels decreased with the aclonifen dose, being significantly lower in the V1, V2, V3, and V4 experimental variants as compared with the control variant (M). The same tendency was identified at the other time points (Table 2). These results provide evidence for an inhibitory effect of aclonifen on UA starting at relatively low exposure times (7 days) and doses (half the recommended dose). There are numerous studies regarding the effect of herbicides on soil UA revealing either no effect [56,57], low negative effect [21], inhibition [14,25,58], or increase in UA [56], depending on the experimental conditions, the properties of herbicides, dose, exposure duration, and soil properties. Soil urease is of microbial origin; it degrades urea in soil, is involved in nitrogen mineralization, and leads to the increase in soil pH [47]. Its inhibition emphasizes changes in soil microbial populations correlated with the delay of urea decomposition and with a possible harmful low local pH value. Herbicide dose had a significant main effect on CA values ($F_{(4, 59)} = 752.8$, $p < 0.001$, $\eta 2 = 0.083$). Post hoc analyses using the Bonferroni approach revealed that the CA values decreased in a dose-dependent manner from the V2 variant onward (Figure 3a). In addition, a significant main effect of exposure duration on CA levels was identified ($F_{(3, 59)} = 1060.92$, $p < 0.001$, $\eta 2 = 0.889$). In contrast to DHA and UA, the impact of this variable was stronger than that of the herbicide dose; that is, it was 10-fold higher. Post hoc testing disclosed an unbalanced pattern of CA variation, with the CA values decreasing significantly after 14 days, increasing at 21 days at values significantly below those measured after 7 days before reaching the lowest levels at 28 days (Figure 3b). The meaning of these effects was interpreted as a significant interaction between the herbicide dose and the exposure time ($F_{(12, 59)} = 77.21$, $p < 0.001$, $\eta 2 = 0.025$). After 7 days, there was a decrease in the CA levels in a dose-dependent manner when compared with the CA value for the control sample (Table 2). After 14 days, this effect was observed for V2–V4 experimental variants (Table 2). The dose–response patterns after 21 and 28 days were consistent with those observed after 7 days (Table 2). These data suggest that aclonifen exerts a dose-dependent inhibitory effect on CA starting from 7–14 days of exposure to a dose lying between half the recommended dose (V1) and the current recommended dose (V2). The literature data also emphasize the inhibition of CA by other herbicides [58,59]. Soil catalase is an enzyme found in aerobic microorganisms. It has an important contribution to soil fertility and is responsible for splitting the hydrogen peroxide into water and oxygen [42]. Considering this information, the inhibition of catalase activity by aclonifen can be also explained by the fact that aclonifen produces a decrease in the populations of aerobic microorganisms and changes in the soil's physicochemical properties.

There also was a statistically significant effect of the herbicide dose on PhA ($F_{(4, 59)} = 77.45$, $p < 0.001$, $\eta 2 = 0.240$). The measured values showed a similar trend with that seen for CA, CA having comparable values for the control and V1 variant and decreasing significantly with the dose for the V2–V4 variants (Figure 3c). Exposure duration exerted a significant effect on the PhA ($F_{(3, 59)} = 263.43$, $p < 0.001$, $\eta 2 = 0.614$) and the relative impact of exposure duration on the PhA was three-fold higher compared with that of the herbicide dose. The decrease in PhA values with time was significant, as indicated by the paired post hoc comparisons (Figure 3d).

A significant interaction between aclonifen dose and exposure time was also detected in the case of PhA ($F_{(12, 59)} = 12.21$, $p < 0.001$, $\eta 2 = 0.011$). The pattern of the dose–response effect was very heterogeneous and more irregular than those seen for the other enzymes' activities. After 7 days, significantly decreased PhA levels were detected only for the V4 variant (Table 2). In contrast, significantly elevated levels were found after 14 days for the V1 variant and after 21 days, significantly reduced PhA was identified for the V4 variant (Table 2). However, the measured values after 28 days were found to decrease significantly in a dose-dependent manner. Overall, these findings support that, under the present experimental conditions, PhA is less sensitive to aclonifen compared with DA, UA, and CA. This outcome is also in correlation with other data revealing that PhA was either stimulated or remained unchanged in the presence of some herbicides [17,57]. However, other studies revealed the inhibition of PhA by herbicides [14,25,58]. Soil phosphatases

are produced by both microorganisms and plant roots and have an important role in phosphorus cycling [60]. The weak inhibition of PhA in the presence of aclonifen can be also due to changes in the microbiological populations and soil properties.

The outcomes of this study indicate that the presence of aclonifen in soil leads to decreased activities of soil enzymes, and the decrease is correlated with the aclonifen dose and exposure time. Furthermore, these results emphasize that 28 days is not enough for the recovery of the enzymatic activities after the treatment with aclonifen. The impaired activities of enzymes found in soil due to the presence of pesticides have been also reported in specific literature. There were different registered responses of the soil's enzymatic activities due to the presence of pesticides, strongly correlated with the pesticide doses and exposure times. There are numerous studies in the specific literature dealing with this subject, with several of these study being mentioned here: (i) the insecticide quinalphos inhibited DHA [20]; (ii) acetamiprid, an insecticide, had an important negative influence on soil PhA [10]; (iii) the pesticide pentachlorophenol negatively affected the DHA even at low concentrations and at high concentration also impaired the PhA [15]; (iv) the herbicides chlorsulfuron and MCPB-Na (4-(4-chloro-o-tolyloxy)butyric acid) strongly inhibited UA and DHA [22]; (v) fomesamen, the other diphenyl ether herbicide, had stimulating effects on the PhA and DHA but significantly inhibited UA [61]; (vi)—thiamethoxam and cypermethrin (other insecticides) showed an inhibitory effect on DHA, UA, CA and PhA [19]; (vii) the herbicide oxyfluorfen manifested an inhibitory effect against the DHA, PhA, UA and protease [25]; (viii) the herbicide S-Metolachlor inhibited DHA, PhA, UA and protease activity [14]; (ix) triazole fungicides inhibited the activities of numerous soil enzymes [26,62]. Similar to the outcomes of this study, the reported effects of pesticides were both dose- and time-dependent.

Statistical analysis revealed moderate and strong positive correlations between the DHA and the soil moisture, pH, OM, and ammonium level. In the case of UA, it correlated significantly only with the soil's nitrate content. For CA, a strong positive relationship with the soil organic content was found, as well as moderate positive associations with the soil ammonium level and available phosphorus. PhA also displayed direct moderate correlations with soil organic matter and available phosphorus.

It is generally recognized that the soil's physicochemical parameters strongly influence the activity of soil enzymes. Specific literature emphasizes a positive effect of the soil OM and nitrate contents and pH on the activity of soil enzymes, especially on the DHA, PhA, and UA [14,25,63,64]. The great complexity of these inter-correlations must also be mentioned, as the presence of the aclonifen may affect the physical, chemical, and biological composition of the soil through its effects on the microbial populations and on the activities of enzymes found in soil.

The results obtained through the molecular docking method are in a good correlation with the experimental outcomes and reveal that aclonifen is able to interact with the investigated enzymes. Figure 4 reveals that aclonifen is able to bind to the active sites of dehydrogenase (the binding position being in the same region as the location of the substrate NADPH), phosphatase (the binding position corresponding to the region of binding of the PHR signaling peptide), and catalase 2 (the binding mode corresponding to the position of binding of hem). Others molecular docking studies revealed that the herbicides nicosulfuron and chlorsulfuron were also able to bind to the active site of the *Proteus mirabilis* catalase [28]; the herbicides S-metolachlor [14] and oxyfluorfen [25], and, respectively, the fungicide triticonazole [27] were able to bind to the catalytic sites of the soil dehydrogenases and phosphatases. Aclonifen did not bind to the active site of urease, as shown in Figure 4d. It can be explicated by the fact that the catalytic site of this enzyme is narrow and hydrophilic and there also is a considerable rigidity of the enzyme skeleton in the vicinity of the active site, the catalysis being controlled by the movement of the helix-loop-helix region 300–350 that transformed the catalytic site into a closed conformation [47]. It does not exclude the possibility that aclonifen produces an allosteric inhibition of urease. Similar results were obtained when docking other pesticides

with urease: the herbicide nicosulfuron and the fungicide difenoconazole [21] and the two stereoisomers of the fungicide triticonazole [27].

Enzymes found in soil perform key functions catalyzing numerous reactions assuring the conversion of organic substances and energy connected to life processes of soil microorganisms. Consequently, the inhibition of activity of soil enzymes has consequences on soil equilibrium and fertility. The activity of these enzymes strongly depends on the soil properties and responses of the microbial community to environmental factors, including the presence of herbicides. Specific literature data usually revealed that herbicide application produced a negative effect on the soil's microbial metabolic activity, microbial density and diversity in parallel with the decrease in soil enzymes' activities. Even if herbicides are applied in the recommended doses, they may cause slight changes in soil microbiota and metabolic activities, but in the long term, herbicide application may cause serious changes in the biochemical balance of the soil, which lead to decreased enzyme activities, soil fertility, and productivity. Consequently, the enzyme assays are able to inform about the level of degradation and/or recovery of soil characteristics. Accordingly, the outcomes of this study should be used to guide the management of aclonifen in correlation with soil properties for long-term sustainability. Furthermore, a proper management of aclonifen use must be considered with respect to recommended application rates and intervals.

Data presented in Table 4 show that aclonifen and its metabolite 4-(3-amino-2-chloro-4-nitrophenoxy)phenol have a moderate aquatic toxicity, with the other metabolites usually revealing a low acute aquatic toxicity. The least toxic of the aclonifen metabolites is revealed by 3,4-diamino-2-chlorophenol. To our knowledge, this is the first study on the toxicity of aclonifen metabolites to aquatic organisms. Furthermore, data presented in Table 5 show that all the soil metabolites of aclonifen are capable of binding to the catalytic sites of phosphatase, dehydrogenase, and catalase 2, with the highest binding energies corresponding to catalase 2, followed by those corresponding to dehydrogenase and to phosphatase, respectively. Moreover, 4-(3-amino-2-chloro-4-nitrophenoxy)phenol is the metabolite revealing higher binding energies and 3,4-diamino-2-chlorophenol is the metabolite emphasizing the lowest binding energies. Similarly, the 4-(3-amino-2-chloro-4-nitrophenoxy)phenol metabolite also revealed the highest toxicity and the 3,4-diamino-2-chlorophenol emphasized the lowest toxicity against aquatic organisms. These outcomes reveal that, among the soil metabolites of aclonifen, the 4-(3-amino-2-chloro-4-nitrophenoxy)phenol emphasized the highest negative effects for both soil enzymes and aquatic organisms. These results emphasize that eco-toxicity tests should be performed not only for pesticides, but also for their metabolites, which may also exert toxicity against numerous aquatic and soil organisms.

## 5. Conclusions

The results obtained in the current study show a significant inhibitory effect of the herbicide aclonifen on the activities of urease, dehydrogenase, phosphatase, and catalase. The most sensitive enzyme was dehydrogenase and the least sensitive was phosphatase to both the herbicide dose and exposure duration. The inhibitory effects on the activity of these enzymes increased with the applied dose of aclonifen, being significant for doses starting with the recommended dose and higher than this. The inhibitory effects on enzymes' activities also depended on the incubation period. Urease activity decreased significantly after 7 days, dehydrogenase activity decreased over a period of 21 days, and phosphatase and catalase activities decreased over the entire incubation period of 28 days. In addition, 28 days was not sufficient to recover the activities of these enzymes. The use of lower doses (half the recommended dose and the recommended dose) of aclonifen resulted in minimal disruption of the activities of enzymes found in soil. Based on the statistical analyses, significant correlations were established between the analyzed enzyme activities and some physicochemical parameters of the soil. The most important soil parameters that influenced enzyme activities were organic matter, nitrogen level, and available phosphorus content.

Data obtained through computational approaches showed that the soil metabolites of aclonifen are able to bind to the investigated enzymes and, consequently, can inhibit their activities. Both aclonifen and its soil metabolites show different degrees of acute toxicity to aquatic organisms. The metabolite 4-(3-amino-2-chloro-4-nitrophenoxy)phenol revealed an acute aqueous toxicity similar to that of aclonifen. This metabolite also revealed the highest capacity to bind to the enzymes found in the soil, its interaction energies with the enzymes being comparable to those of aclonifen.

**Supplementary Materials:** The following supporting information can be downloaded at: https://www.mdpi.com/article/10.3390/agriculture13061226/s1, Figure S1: Superposition of the structures of the catalytic domains the Bacillus subtillis catalase 2 (of AlphaFold model AF-P42234-F1, red ribbon) and the Exiguobacterium oxidotolerans catalase (PDB ID 2J2M chain A, dark blue ribbon). Supplementary Table S1: Physicochemical parameters of the soil.

**Author Contributions:** All authors designed the study, analyzed and interpreted the results, and revised the manuscript; M.N.C. performed biological experiments and collected data; I.V.C. collected data and performed statistical analysis; D.L.R. and A.I. performed the bioinformatics study and collected data; D.L.R. and A.I. edited the manuscript. All authors have read and agreed to the published version of the manuscript.

**Funding:** This research received no external funding.

**Institutional Review Board Statement:** Not applicable.

**Data Availability Statement:** Not applicable.

**Acknowledgments:** We would like to thank the University of Life Sciences "King Mihai I" from Timisoara for the financial support granted for the publication of the article.

**Conflicts of Interest:** The authors declare no conflict of interest.

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
