# Peer review of "Assessment of the Effects of the Herbicide Aclonifen and Its Soil Metabolites on Soil and Aquatic Environments"

_agriculture, doi:10.3390/agriculture13061226_

Round 1

Reviewer 1 Report

After reviewing the present manuscript, reviewer found that the authors made interesting work. The reviewer found that this manuscript is fit with Journal "Agriculture".

 Additional comments:

1. What is the main question addressed by the research?

      This study aims  assess the effects of aclonifen and of its known soil metabolites on the enzymes found in soil.

 2. Do you consider the topic original or relevant in the field? Does it address a specific gap in the field?

    Topic is interesing. The novelty should be stressed in the Introduction section.

 3. What specific improvements should the authors consider regarding the manuscript? What further controls should be considered?

In Figures 2 and 3, simplify the description of the Y-axis. It is enough to specify only the unit of a given enzyme, because its name is in the title of the figure.

I suggest dividing the Results and Discussion chapter into two separate ones: Results, in which you briefly describe the results obtained, and the Discussion chapter, in which you justify your results in the context of the latest available literature. In this chapter also state the limitations of your research. This solution will greatly strengthen your manuscript.

 4. Are the conclusions consistent with the evidence and arguments presented and do they address the main question posed?

    Yes, conclusions are satisfactory with the addressed issue.

 5. Are the references appropriate?

    Yes, reference are satisfied. Make sure all items are cited in the text and in the table of contents and vice versa. In the chapter, please add doi.

Author Response

Thank you very much for the review and for the suggestions that are clearly meant to improve the quality of our manuscript. In what follows, we answer the requirements and hope that we have understood them correctly. Our text is in blue.

  1. 1. What is the main question addressed by the research?

      This study aims assess the effects of aclonifen and of its known soil metabolites on the enzymes found in soil.

  1. Do you consider the topic original or relevant in the field? Does it address a specific gap in the field?

    Topic is interesting. The novelty should be stressed in the Introduction section.

We have modified the last paragraph in the Introduction section such as to emphasize the novelty of the study (new lines 26-29).

  1. 3. What specific improvements should the authors consider regarding the manuscript? What further controls should be considered?

In Figures 2 and 3, simplify the description of the Y-axis. It is enough to specify only the unit of a given enzyme, because its name is in the title of the figure.

The Figures 2 and 3 are simplified, there are new figures inserted in the manuscript.

I suggest dividing the Results and Discussion chapter into two separate ones: Results, in which you briefly describe the results obtained, and the Discussion chapter, in which you justify your results in the context of the latest available literature. In this chapter also state the limitations of your research. This solution will greatly strengthen your manuscript.

Thank you for this suggestion also. The Results and Discussions chapter has been divided as suggested.

  1. Are the conclusions consistent with the evidence and arguments presented and do they address the main question posed?

    Yes, conclusions are satisfactory with the addressed issue.

  1. Are the references appropriate?

    Yes, references are satisfied. Make sure all items are cited in the text and in the table of contents and vice versa. In the chapter, please add doi.

The DOI are added for all references where it applies.

Reviewer 2 Report

General comment:

The manuscript entitled “Assessment of the effects of the herbicide aclonifen and its soil metabolites on soil and aquatic environment” is well written and structured research work and an interesting topic for readers from various fields including agronomy, pesticide science, environmental science, etc. The following revisions need to be considered before publication:

-Abstract: Readers often evaluate studies based on abstracts, as you know. Readers should be able to determine the robustness of each study's main findings from abstracts that provide enough statistical data. It is therefore important to provide some key statistical results in the abstract.

-Introduction: Pesticides are the main topic of the data and literature used in the introduction! In particular, I recommend concentrating on herbicides and Aclonifen.

-Introduction, Lines 29-32: Cite appropriately, for example:

Mehdizadeh et al. 2021:

Mehdizadeh, M.; Mushtaq, W.; Anusha Siddiqui, S.; Ayadi, S.; Kaur, P.; Yeboah, S.; Mazraedoost, S.; AL-Taey, D.K.A.; Tampubolon, K. Herbicide Residues in Agroecosystems: Fate, Detection, and Effect on Non-Target Plants. Rev. Agric. Sci. 2021, 9, 157–167.

Mehdizadeh, 2016:

Mehdizadeh M. 2016. Effect of pesticide residues on agricultural food production; a case study: sensitivity of oilseed rape to Triasulfuron herbicide soil residue. MOJ Food Process Technol 2:1-2. https://doi.org/10.15406/mojfpt.2016.02.00054

-Introduction, Lines 45-47: Vague sentences! It can be revised as follows:

"Thus, soil microorganism populations and enzyme activities, in combination with the size and biodiversity of soil microorganisms, can be used as bio-indicators of changes to soil environments caused by pesticides."

-Introduction, Lines 111-113: This information is not necessary. It should be removed.

- Materials and Methods, Lines 120-122: The sentences are vague. It needs to be revised.

-Results and Discussion: There is a good presentation of the results section, and the use of informative figures and tables enriches it. However, the Discussion part of this research is very weak and not sufficient. Discussions serve as a means of interpreting and describing your findings in light of what you already know about the research problem under investigation, as well as explaining any new insight or understanding you have gained after taking the findings into account. So, it is necessary to improve the discussion on a fundamental level.

-Results and Discussion, Lines 264-265: Make sure these values are correct.

Due to the presence of some vague sentences in the text, the language needs to be reviewed by an expert. The attached file contains some of these vague sentences.

Author Response

Thank you very much for the review and for the suggestions that are clearly meant to improve the quality of our manuscript. In what follows, we answer the requirements and hope that we have understood them correctly. Our text is in green.

General comment:

The manuscript entitled “Assessment of the effects of the herbicide aclonifen and its soil metabolites on soil and aquatic environment” is well written and structured research work and an interesting topic for readers from various fields including agronomy, pesticide science, environmental science, etc. The following revisions need to be considered before publication:

-Abstract: Readers often evaluate studies based on abstracts, as you know. Readers should be able to determine the robustness of each study's main findings from abstracts that provide enough statistical data. It is therefore important to provide some key statistical results in the abstract.

A new sentence has been introduced in the abstract to stress the statistical significance of the results (lines 22-25).

-Introduction: Pesticides are the main topic of the data and literature used in the introduction! In particular, I recommend concentrating on herbicides and Aclonifen.

The old lines 33-37 have been eliminated and the corresponding references deleted. Anyway, information regarding the statistics on the production and use of pesticides and on the regulations of the pesticides use in soil and water have been introduced as it was asked by another reviewer.

-Introduction, Lines 29-32: Cite appropriately, for example:

Mehdizadeh et al. 2021:

Mehdizadeh, M.; Mushtaq, W.; Anusha Siddiqui, S.; Ayadi, S.; Kaur, P.; Yeboah, S.; Mazraedoost, S.; AL-Taey, D.K.A.; Tampubolon, K. Herbicide Residues in Agroecosystems: Fate, Detection, and Effect on Non-Target Plants. Rev. Agric. Sci. 2021, 9, 157–167.

Mehdizadeh, 2016:

Mehdizadeh M. 2016. Effect of pesticide residues on agricultural food production; a case study: sensitivity of oilseed rape to Triasulfuron herbicide soil residue. MOJ Food Process Technol 2:1-2. https://doi.org/10.15406/mojfpt.2016.02.00054

The references Mehdizadeh, 2016 and Mehdizadeh et al. 2021 have been added, they have numbers 2 and 3, new lines 56-58.

-Introduction, Lines 45-47: Vague sentences! It can be revised as follows:

"Thus, soil microorganism populations and enzyme activities, in combination with the size and biodiversity of soil microorganisms, can be used as bio-indicators of changes to soil environments caused by pesticides."

This replacement is done, new lines 60-65.

-Introduction, Lines 111-113: This information is not necessary. It should be removed.

The information has been removed.

- Materials and Methods, Lines 120-122: The sentences are vague. It needs to be revised.

The sentences ware corrected, lines 150-156.

-Results and Discussion: There is a good presentation of the results section, and the use of informative figures and tables enriches it. However, the Discussion part of this research is very weak and not sufficient. Discussions serve as a means of interpreting and describing your findings in light of what you already know about the research problem under investigation, as well as explaining any new insight or understanding you have gained after taking the findings into account. So, it is necessary to improve the discussion on a fundamental level.

The Results and Discussion chapter has been divided as requested by other Reviewer. The discussion was enlarged tacking into account your suggestions, thank you very much.

-Results and Discussion, Lines 264-265: Make sure these values are correct.

We checked the values, they are correct.

Comments on the Quality of English Language

Due to the presence of some vague sentences in the text, the language needs to be reviewed by an expert. The attached file contains some of these vague sentences.

The English has been checked and we did our best to correct the sentences. The corrections suggested by the Reviewer have been considered.

Reviewer 3 Report

The manuscript titled: “Assessment of the effects of the herbicide aclonifen and its soil metabolites on soil and aquatic environment” - agriculture-2412537 is quite interesting.

In the studies, the authors determined the toxicity of aclonifen and its soil metabolites on aquatic organisms and the interaction of these compounds with soil enzymes.

Unfortunately, in my opinion, in order for the manuscript to be published in this prestigious journal, its content needs to be changed and improved, especially the introduction chapter. The research methods and results are well described. The statistical analysis of the data is presented at an acceptable level. The discussion needs to be expanded and the conclusions need to be reworded.

GENERAL COMMENTS

Abstract

“… (dehydrogenases, phosphatases, ureases and catalases) have been evaluated 14 in laboratory conditions for soil treated with various doses of aclonifen at every 7 days for a period 15 of 28 days. A computational approach was considered to assess the acute toxicity of aclonifen and 16 of its soil metabolites on aqueous organisms and the interactions of these compounds with soil en- 17 zymes. The results obtained in the experimental study have shown a significant inhibitory effect of 18 the herbicide aclonifen on the investigated enzymes, those activities were not recovered after 28 19 days in the presence of high doses of aclonifen. The investigated enzyme activities were strongly 20 influenced by the organic matter, nitrogen level and available phosphorus content. The outcomes 21 of the computational study revealed moderate acute aqueous toxicity of aclonifen and of its metab- 22 olites. All metabolites were also able to bind to the investigated enzymes and may have an inhibitory 23 effect on their activities.

What phosphatases? Acid or alkaline phosphatase? .....probably one urease, and catalase?

“…for soil treated with various doses of aclonifen at every 7 days for a period of 28 days.”

What doses?

“A computational approach was considered to assess the acute toxicity of aclonifen and of its soil metabolites on aqueous organisms and the interactions of these compounds with soil enzymes.”

Considered or made calculations?

Introduction

·        Please provide statistics on the production and use of pesticides. What legal regulations constitute their use in soil and water.

·        “The effects of pesticides on the soil are mainly due to the fact that they can affect the soil microbial populations by influencing the survival and reproduction of individual species. Furthermore, the activities of enzymes found in soil are strongly correlated with the populations of soil microorganisms. Consequently, the size and biodiversity of the population of soil microorganisms, added to the activities of enzymes found in soil, can serve as bio-indicators of the changes occurring in the soil environment due to pesticides application [3].”

This manuscript excerpt does not add to its value. Instead, it would be good to describe the ways in which pesticides are degraded in soils.

·        “Different groups of microorganisms found in soil are able to degrade pesticides, fungi and bacteria being the groups mainly involved into the degradation and mineralization of the pesticides through enzymatic reactions [4-7].”

What enzymatic reactions and what microbes? Please name at least a few key microorganisms.

·        Urease and catalase should be given in the singular.

·        “The global aclonifen market is considered to grow at an annual growth rate of 6.48% during the period 2021-2027 [34]. Consequently, assessment of its effects on soil and aquatic environments are more than necessary. From the experimental point of view, the activity of several enzymes (dehydrogenase, urease, phosphatase and catalase) used as soil quality indicators and the physicochemical properties of soil are assessed over a period of 28 days after the application of various doses of aclonifen, in laboratory conditions. Computationally, the effects of the aclonifen and of its known soil metabolites on the aquatic organisms are predicted and the molecular docking approach has been considered to evaluate the interactions of both aclonifen and of its metabolites with specific enzymes found in soil.

This part of the chapter should appear above the research objective. It would be good if the goal and Hypothesis were described as a consequence of the data.”

This part of the chapter should appear above the research objective. It would be good if the goal and Hypothesis were described as a consequence of the data.

·        “To the best of our knowledge, this is the first study dealing with the effects of aclonifen and of its soil metabolites on the activities of enzymes found in soil and on the aquatic organisms. A simple search about “aclonifen” in Science Direct scientific database yields only 277 results for all years, but none of them deals with issues similar to those presented in this study.”

This sentence should be reworded, the authors want too much to prove that the research is unusual within the Science Direct platform.

Conclusion

·        Which enzyme was the most sensitive, and which was the least?

·        What specific dose in mg kg-1 d.m. soil was toxic? This does not follow from the conclusion.

In table 5 there should be dehydrogenases or enzyme abbreviations.

Author Response

Thank you very much for the review and for the suggestions that are clearly meant to improve the quality of our manuscript. In what follows, we answer the requirements and hope that we have understood them correctly. Our text is in purple.

The manuscript titled: “Assessment of the effects of the herbicide aclonifen and its soil metabolites on soil and aquatic environment” - agriculture-2412537 is quite interesting.

In the studies, the authors determined the toxicity of aclonifen and its soil metabolites on aquatic organisms and the interaction of these compounds with soil enzymes.

Unfortunately, in my opinion, in order for the manuscript to be published in this prestigious journal, its content needs to be changed and improved, especially the introduction chapter. The research methods and results are well described. The statistical analysis of the data is presented at an acceptable level. The discussion needs to be expanded and the conclusions need to be reworded.

 GENERAL COMMENTS

Abstract

“… (dehydrogenases, phosphatases, ureases and catalases) have been evaluated 14 in laboratory conditions for soil treated with various doses of aclonifen at every 7 days for a period 15 of 28 days. A computational approach was considered to assess the acute toxicity of aclonifen and 16 of its soil metabolites on aqueous organisms and the interactions of these compounds with soil en- 17 zymes. The results obtained in the experimental study have shown a significant inhibitory effect of 18 the herbicide aclonifen on the investigated enzymes, those activities were not recovered after 28 19 days in the presence of high doses of aclonifen. The investigated enzyme activities were strongly 20 influenced by the organic matter, nitrogen level and available phosphorus content. The outcomes 21 of the computational study revealed moderate acute aqueous toxicity of aclonifen and of its metab- 22 olites. All metabolites were also able to bind to the investigated enzymes and may have an inhibitory 23 effect on their activities.

What phosphatases? Acid or alkaline phosphatase? .....probably one urease, and catalase?

 The text has been corrected.

“…for soil treated with various doses of aclonifen at every 7 days for a period of 28 days.”

What doses?

The used doses have been added in the abstract.

“A computational approach was considered to assess the acute toxicity of aclonifen and of its soil metabolites on aqueous organisms and the interactions of these compounds with soil enzymes.”

Considered or made calculations?

 The text has been modified such as to reflect that the acute toxicity has been computed and the interactions with enzymes have been assessed.

Introduction

  • Please provide statistics on the production and use of pesticides. What legal regulations constitute their use in soil and water.

The requested information has been added, new lines 35-45.

  • “The effects of pesticides on the soil are mainly due to the fact that they can affect the soil microbial populations by influencing the survival and reproduction of individual species. Furthermore, the activities of enzymes found in soil are strongly correlated with the populations of soil microorganisms. Consequently, the size and biodiversity of the population of soil microorganisms, added to the activities of enzymes found in soil, can serve as bio-indicators of the changes occurring in the soil environment due to pesticides application [3].”

This manuscript excerpt does not add to its value. Instead, it would be good to describe the ways in which pesticides are degraded in soils.

The requested information has been added, lines 63-69.

“Different groups of microorganisms found in soil are able to degrade pesticides, fungi and bacteria being the groups mainly involved into the degradation and mineralization of the pesticides through enzymatic reactions [4-7].”

What enzymatic reactions and what microbes? Please name at least a few key microorganisms.

The requested information has been added, lines 75-77.

  • Urease and catalase should be given in the singular.

These terms are corrected over the manuscript.

  • “The global aclonifen market is considered to grow at an annual growth rate of 6.48% during the period 2021-2027 [34]. Consequently, assessment of its effects on soil and aquatic environments are more than necessary. From the experimental point of view, the activity of several enzymes (dehydrogenase, urease, phosphatase and catalase) used as soil quality indicators and the physicochemical properties of soil are assessed over a period of 28 days after the application of various doses of aclonifen, in laboratory conditions. Computationally, the effects of the aclonifen and of its known soil metabolites on the aquatic organisms are predicted and the molecular docking approach has been considered to evaluate the interactions of both aclonifen and of its metabolites with specific enzymes found in soil.

This part of the chapter should appear above the research objective. It would be good if the goal and Hypothesis were described as a consequence of the data.”

 This paragraph has been moved before the research objectives.

  • “To the best of our knowledge, this is the first study dealing with the effects of aclonifen and of its soil metabolites on the activities of enzymes found in soil and on the aquatic organisms. A simple search about “aclonifen” in Science Direct scientific database yields only 277 results for all years, but none of them deals with issues similar to those presented in this study.”

 This sentence should be reworded, the authors want too much to prove that the research is unusual within the Science Direct platform.

The paragraph has been changed. One sentence has been removed and the remaining one emphasize the novelty of the study, as another Reviewer asked (lines 137-143).

Conclusion

  • Which enzyme was the most sensitive, and which was the least?
  • What specific dose in mg kg-1d.m. soil was toxic? This does not follow from the conclusion.

 This information has been added in the Conclusions chapter.

 In table 5 there should be dehydrogenases or enzyme abbreviations.

It is modified.  

Round 2

Reviewer 2 Report

The authors have addressed all my concerns in the revised version of the manuscript. Accordingly, I believe this manuscript could be accepted in this format.

The authors have revised the vague sentences in the revised file.